# Detection of *Enterobius vermicularis* in archived formalin-fixed paraffin-embedded (FFPE) appendectomy blocks: It's potential to compare genetic variations based on mitochondrial DNA (*cox*1) gene

Maryam Haghshenas[1], Mona Koosha[2], Alireza Latifi[1], Elham Kazemirad[1]*, Arash Dehghan[3], Bahram Nikmanesh[4], Gholamreza Mowlavi[1,5]*

1 Department of Medical Parasitology and Mycology, School of Public Health, Tehran University of Medical Sciences, Tehran, Iran, 2 Department of Medical Entomology and Vector Control, School of Public Health, Tehran University of Medical Sciences, Tehran, Iran, 3 Department of Pathology, School of Medicine, Hamedan University of Medical Sciences, Hamedan, Iran, 4 Department of Medical Laboratory Sciences, School of Allied Medical Sciences, Tehran University of Medical Sciences, Tehran, Iran, 5 Center for Research of Endemic Parasites of Iran (CREPI), Tehran University of Medical Sciences, Tehran, Iran

* kazemirad@tums.ac.ir, ekazemirad@yahoo.com (EK); molavig@yahoo.com (GM)

**Data Availability Statement:** All relevant data are within the paper and its Supporting Information

## Abstract

Acute appendicitis represents one of the most common causes of emergency abdominal surgery worldwide. Meanwhile, *Enterobius vermicularis* has been suggested as one of the probable causes of appendicitis. In this study, the morphological characteristics of the remnant pinworms and pathologic changes were explored in old-archived FFPE tissues of appendectomies. Moreover, we provide the first molecular identification, genetic, and haplotype variation of this nematode from the old-archived FFPE tissue section of appendectomy using the mitochondrial cytochrome c oxidase subunit 1 (*cox*1) gene. Seventeen FFPE appendectomies with *E. vermicularis* infection, stored over 12–22 years, were collected from two different geographical areas of Iran. In the histopathological examination, tissue changes were observed in thirteen cases (76.4%) and inflammation in four blocks (23.5%). After DNA extraction, the *cox*1 gene was amplified in twelve (70.6%) cases using the nested polymerase chain reaction (PCR). Phylogenetic analysis and a median-joining network of 78 available *cox*1 sequences of *E. vermicularis* revealed 59 haplotypes. We identified five haplotypes that fell into type B. All Haplotypes are novel except for two haplotypes, Hap32 and Hap37, identical to *E. vermicularis* sequences from Iran, Greece, and Germany. The ranges of diversity distance and haplotype diversity within the isolates were 0–1.9% and HD:0.643–0.667, subsequently. Overall, the absence of inflammation or even tissue changes in some sections can suggest the possible non-inflammatory role of *E. vermicularis* in appendicitis. Although FFPE material suffers from PCR inhibition, we could successfully use nested PCR to characterize *E. vermicularis* in old-archived appendectomy blocks and suggest this method as a complementary diagnosis technique in pathology. While the predominant type was B in the Middle East and Europe, further studies on a larger sample size

files. All sequences are available from the GenBank database (accession number(s) MZ361991-9, MZ360956-8).

**Funding:** This research has been supported by Tehran University of Medical Sciences & health Services grant no. 1400-1-211-52257. The funders had no role in study design, data collection, and analysis, decision to publish, or preparation of the manuscript.

**Competing interests:** The authors have declared that no competing interests exist.

from different geographical regions could probably confirm the results obtained in the present study.

## Introduction

*Enterobius vermicularis* is one of the most common parasitic helminths distributed worldwide, with prevalence rates of 30–50% [1], which is considered a possible causative agent of acute appendicitis [2]. The parasite mostly affected children aged 5–10 years, accounting for more than 30% of all infected cases [3]. Typically, the large intestine is the primary habitat of the adult females of *E. vermicularis*, and the infection transmission occurs through picking up the eggs by the fingers during the scratching due to perianal pruritus. However, their migration to the perianal skin for laying eggs during the night can facilitate retroreflection and autoinfection by ingestion or inhalation of eggs [3]. Meanwhile, the prevalence rate of infection has been recorded to be high in institutional populations and within family contacts [4].

It should be noted that most cases are asymptomatic in almost 40% of infected ones [5]; however, some rare and unusual cases of invasive *E. vermicularis* infection in kidneys and fallopian tubes can be lethal [6]. Regardless of the high-risk prevalent clinical manifestations in childhood that some might be assumed to be associated with the incidence of parasitic infections, acute appendicitis due to enterobiasis has remained controversial [2, 7].

At the beginning of the 21st century, the incidence of appendicitis was seen to grow in newly industrialized countries. Moreover, mortality, morbidities, and the imposed economic impact associated with appendicitis have been reported in western countries [8]. Fits described appendicitis first-time more than 100 years ago, and 561,000 appendectomies per year were conducted between 1979–1984 in the United States, representing that different etiological agents might have had hands in this regard [9]. Meanwhile, factors like low fiber diet, hygiene, ethnic and genetic features, socioeconomic status, seasonal variation, temperature, and altitude, along with bacterial, viral, and parasitic infections, have influenced appendicitis [4]. With an overview of the parasitic worms mentioned as possible etiological agents in appendicitis, the name *E. vermicularis* is the most highlighted, with a more special incidence in children and young adults [10].

In the last decades, limited target genes have been used to explore the genetic diversity, phylogeography, and host specificity of *E. vermicularis*. Mitochondrial DNA (mtDNA) sequence data has been used as an appropriate genetic marker for population genetic studies [11]. Cytochrome c oxidase subunit 1 (*cox*1) gene is one of the most diverse mitochondrial genes used for the phylogenetic analysis of *E. vermicularis* [12–14]. It can provide a more comprehensive representation of *E. vermicularis* evolution than other genes like ribosomal DNA. To date, *cox*1 sequences of *E. vermicularis* have been published in only a few countries. The analysis of the mitochondrial *cox*1 gene sequences in eggs and adult worms from humans and captive chimpanzees (*Pan troglodytes*) in Japan has shown the presence of three different genetic types, designated as type A, B, and C [15]; type A has been identified in both species, whereas types B and C only in chimpanzees. In contrast, all the isolates were grouped in type B based on the cox1 sequencing of *E. vermicularis* eggs from humans in Greece, Denmark, Poland, Germany, and Iran [16–18]. In a recent study in Thailand, types A and B of *E. vermicularis* were isolated from humans by an adhesive tape perianal swab technique, suggesting that type A might be restricted to eastern Asia [19].

To date, no studies have been conducted on FFPE tissue from appendectomy to explore molecular characterization and genetic variation of *E. vermicularis*. The FFPE tissues are

valuable resources preserving the tissues with intact morphological and tissue structures. However, due to the high fragmentation of DNA, successful amplification of DNA is quite challenging, and it needs shorter amplification products of 250 to 300 bp in length or higher sensitive techniques such as nested PCR [20, 21]. In the present study, from the etiological point of view, nested PCR was conducted on the *cox*1 gene to confirm the presence of *E. vermicularis* in old-archived FFPE appendectomy samples. Moreover, we tried to use a higher amplicon length of up to 378 bp to characterize the possible genomic and haplotype variations among all the deposited *cox*1 sequences in GenBank. Furthermore, we explored the morphological characteristics of the remnant pinworms and pathological changes in archived appendectomy samples belonging to two different geographical areas in Iran.

## Materials and methods

### Sample collection

This study was approved by the Ethical Committee of the Tehran University of Medical Sciences, Iran (ethical code: IR.TUMS.SPH.REC.1398.183). The referred ethics committee waived the need for consent to utilize tissue samples. We used 17 formalin-fixed paraffin-embedded (FFPE) samples, 14 blocks from Besat Hospital in Hamadan, western Iran, and three blocks from Tehran Medical Center Hospital, Iran, by reviewing the archived files of patients with appendicitis caused by *E. vermicularis* (Table 1). The FFPE appendectomy blocks had been archived for 12–22 years. The initial parasitological confirmation of *E. vermicularis* was based on detecting the eggs or the remaining parts of the worms in the tissue sections. Demographic and clinical features were regarded in the medical records, while the gross and histological findings were also concerned.

### Histotechniqucal procedure

Tissue processing and standard Hematoxylin and eosin staining (H&E) were performed for histopathological analysis [22]. After identifying a parasitic structure in the stained section, the pathological appearances were determined for each case. An adult form of *E. vermicularis* preserved in 70% alcohol that belonged to a patient from Mazandaran province, northern Iran, was also used as a standard positive sample. The FFPE tissue samples were sectioned between 5 to 10 μm thickness using a microtome, afterward, stained with hematoxylin-eosin (H&E), and mounted on a glass slide for microscopic observation. Finally, the morphological and pathological features of the sections were checked using light microscopy at different magnifications. Diagnosis of *Enterobius vermicularis* in the appendix cross-section was based on the size, narrow lateral cuticular alae, and the prominent platymyarian and meromyarian somatic muscles, a musculature consisting of two to three muscle layers per quarter section divided by four cords [23].

### DNA extraction

DNA extraction was performed from seventeen old-archived formalin-fixed paraffin-embedded (FFPE) tissue samples (Table 1). Three serial sections of 8–10 μm thickness were cut from each block using a microtome (Leica SM2000 R Sliding Microtome, Wetzlar, Germany). The tissue sections were placed on the slides, incubated at 56˚C for 30 minutes, and then soaked for 10 minutes in xylene for deparaffinization. The sections were rehydrated by incubation in a descending graded ethanol series including 100%, 80%, 60%, and 40%, and finally, double distilled water for 10 s each. Tissue sections were dried at room temperature for 1 hour, and subsequently, all three sections of each sample were scraped with a sterile scalpel and

**Table 1. Description of appendectomy sections based on age, pathological, and molecular results.**

| Number of blocks | Block Codes | Relevant hospital, province | Block history | Gender | Age of patient | Pathological outcome | | Nested PCR cox1 | Accession No |
|---|---|---|---|---|---|---|---|---|---|
| | | | | | | Inflammation | Tissue changes | | |
| 1 | 77–648 | Besat Hospital, Hamedan | 1998 | Female | 18 | - | + | + | MZ361991 |
| 2 | 77–649 | Besat Hospital, Hamedan | 1998 | Female | 13 | - | - | + | MZ361992 |
| 3 | 79–475 | Besat Hospital, Hamedan | 2000 | Male | 14 | + | + | + | MZ361993 |
| 4 | 77–846 | Besat Hospital, Hamedan | 1998 | Male | 1 | + | + | + | MZ361994 |
| 5 | 77–143 | Besat Hospital, Hamedan | 1998 | Female | 30 | - | + | + | MZ361995 |
| 6 | 77–667 | Besat Hospital, Hamedan | 1998 | Male | 11 | - | - | - | - |
| 7 | 79–478 | Besat Hospital, Hamedan | 2000 | Male | 6 | + | + | + | MZ361996 |
| 8 | 79–367 | Besat Hospital, Hamedan | 2000 | Female | 20 | - | + | - | - |
| 9 | 77–483 | Besat Hospital, Hamedan | 1998 | Female | 35 | - | + | + | MZ361997 |
| 10 | 79–339 | Besat Hospital, Hamedan | 2000 | Male | 23 | + | + | - | - |
| 11 | 79–535 | Besat Hospital, Hamedan | 2000 | Female | 19 | - | + | - | - |
| 12 | 77–653 | Besat Hospital, Hamedan | 1998 | Female | 25 | - | - | - | - |
| 13 | 83–2331 | Tehran Medical Center | 2004 | Male | 11 | - | + | + | MZ360956 |
| 14 | 77–654 | Besat Hospital, Hamedan | 1998 | Male | 25 | - | + | + | MZ361998 |
| 15 | 77–672 | Besat Hospital, Hamedan | 1998 | Female | 35 | - | - | + | MZ361999 |
| 16 | 85–778 | Tehran Medical Center | 2006 | Female | 5.5 | - | + | + | MZ360957 |
| 17 | 87–2013 | Tehran Medical Center | 2008 | Female | 11 | - | + | + | MZ360958 |
| Total | | | | Female 58.8% Male 41.1% | | 23.5% | 76.4% | 70.5% | |

transferred into a 1.5 ml microcentrifuge tube. Afterward, a High Pure PCR template preparation kit (Roche, Germany) was used for DNA extraction. Briefly, the scraped tissue was digested with 200 μl tissue lysis buffer and 40 μl proteinase k (20 mg/ml) at 56°C for 1 hour. The samples were incubated with 200 μl binding buffer for an additional 10 min at 70°C. After adding 200 μl isopropanol, the lysed emulsion was washed and purified using a collection filter tube according to the manufacturer's instructions. The final elution volume was 60 μl, and all extracted DNA was stored at -20°C.

## Nested PCR method

The partial mitochondrial cytochrome c oxidase 1 (*cox*1) gene of *E. vermicularis* was amplified by a nested PCR method using the outer primers CO1F (5′-TGGTTTTTTGTGCATCCT-GAGGTTTA-3′), CO1R (5′-AGAAAGAACGTAATGAAAATGAGCAAC-3′) [24], and inner

primers EVIF (5–`TTGGTCATCCTGAGGTTTATATTC`–3), EVIR (5–`TCCAAAATAGGAT-`
`TAGCCAACA`–3) [25]. All cycling conditions were as follows: initial denaturation at 94˚C for 6
min, followed by 45 cycles of 94˚C for 1 min, 58.2˚C for 1 min, and 72˚C for 1 min; and a final
elongation step at 72˚C for 10 min. Two μl of the outer PCR reaction were used as a template
for the inner PCR. The conditions for the inner PCR were the same as those for the outer PCR,
except that the annealing temperature was set at 50.1˚C, and the number of cycles was reduced
to 35.

PCR inhibition was reduced by increasing the polymerase concentration, dNTP concentration, and PCR elongation time. Positive control containing DNA extracted from an *E. vermicularis* worm sample and negative control containing DNA-free water were included in all
runs. In addition, appendectomy tissue related to other pathogens was used as the negative
control to check the specificity of primers.

PCR products were separated with 1.5% agarose gel electrophoresis containing Green
Viewer and were photographed using the Vilber Lourmat gel documentation system. Finally,
the PCR products were sequenced bidirectionally via Genetic Codon Company, Tehran, Iran,
using the inner primers set (EVIF/R).

## Analysis of sequence and population genetics

The quality of crude sequences was improved using the Chromas 2.6.6 program by removing
areas with poor quality at both ends of the sequences. The consensus of confident sequences
was examined using the NCBI (nucleotide collection) database. Multiple DNA sequences were
aligned using the ClustalW program (http://www.ebi.ac.uk/clustalw/) and afterward trimmed.
Finally, a 333 bp consensus sequence length was used for phylogenetic analysis. Relationships
were analyzed by Maximum Likelihood, Neighbor-Joining, and UPGMA (unweighted pair
group method with arithmetic mean) inference methods embedded in MEGA X. The best-fit
substitution model for the *cox*1 alignment was the Hasegawa–Kishino–Yano model (HKY+G).
The topology of the phylogenetic tree was evaluated using the bootstrap test based on 1000
replications. For phylogenetic analysis, we used 91 sequences, including 75 *cox*1 sequences of
*E. vermicularis* available in GenBank, 12 sequences from the current study, as well as those of
*Ascaris suum* (accession no. X54253), *Trypanoxyuris atelis* (accession no. AB222177), *Trypanoxyuris microon* (accession no. AB222176) and *Enterobius anthropopitheci* (accession no.
AB254450).

The genetic diversity indices, such as haplotype diversity (HD) and nucleotide diversity (π),
were calculated by using DnaSP version 6.12.03 [26] in the isolates of *E. vermicularis*. Also, an
input file for drawing a median-joining network in PopART software version 1.7 was prepared
[27, 28]. Intra- and interspecific genetic differences were calculated with MEGA version X
[29, 30] by using the Kimura two-parameter (K2P) distance model [29]. Ninety-one *cox*1
sequences (12 sequences from this study, the remainder from Korea, Japan, Thailand, Iran,
Czech Republic, Greece, Germany, Denmark, and Sudan obtained from GenBank) derived
from human and chimpanzee hosts were used in the analysis. The sequences reported in this
paper were deposited in GenBank.

## Results

### Demographic characteristics of appendectomy specimens

The initial histopathological diagnoses of the selected cases were appendicitis due to *E. vermicularis*. Eleven samples (64.71%) were from patients in the age group under 20 years, and six
cases (35.29%) were in patients over 20 years of age. Most appendicitis has occurred in the age
group under 20 years, which could be justified by the prevalence of the disease at younger

ages. Females with ten cases (58.8%) out of seventeen were seen as the predominant gender of appendicitis due to *E. vermicularis* (Table 1). It is worth mentioning that fourteen paraffin-blocked samples (82.3%) were derived from the old archives of 20–22 years ago, and three (17.6%) samples from the past 12–16 years old, while the oldest cases belonged to 1998. (Table 1).

## Pathological findings

Inflammation and tissue changes as histopathological findings were presented in some cases. In contrast, in the rest of the sections, the only microscopical finding was the presence of the worm. Tissue changes were observed in thirteen cases (76.5%) and inflammation in four blocks (23.5%) (Table 1). In nine samples (52.94%), no inflammation was observed in the histopathological analysis despite detecting tissue changes. Meanwhile, it is worth mentioning that only in four samples inflammation and tissue changes were seen together. On the other hand, four cases did not exhibit tissue changes or inflammation, although initially had been diagnosed as acute appendicitis caused by *E. vermicularis*. The absence of inflammation or even tissue changes may drag the minds to the possible non-inflammatory role of *E. vermicularis* in causing acute appendicitis.

The appendix wall can be seen in the histopathological Fig 1A–1C. Lymph follicle hyperplasia with germinal centers in the submucosal layer is apparent. Although the *E. vermicularis* worm was seen inside the appendix lumen, there was no evidence of acute appendicitis, which could be controversial pathologically.

## Microanatomy of *Enterobius vermicularis* in cross-sections

All histopathological slides taken from the hospitals and the newly prepared sections were examined microscopically, and the sections having parasite remnants were drawn using camera lucida at various magnifications. The characteristic morphological appearance of *E. vermicularis* is illustrated in positive slides (Fig 1). Since the male *E. vermicularis* has a shorter life span and may rarely be seen in the intestinal lumen, it seems that the primary morphological identity of the worm sections should have been more focused on females. According to the cross-section diameter, the female *E. vermicularis* is 0.3 to 0.5 mm wide. The cephalic alae are the most common diagnostic sign in the cross-section of *E. vermicularis*, which can be seen bilaterally and symmetrically in the sections (Fig 1). Another taxonomical and specific character is the platymyarian and meromyarian types of the somatic muscle, illustrated in the circular diameter per every four quadrants. In the meromyarian type, the somatic muscles are between two to five in each quadrant in cross-section. Also, platymyarian refers to the flattened basal and sarcoplasmic portions in somatic musculatures. These morphological signs confirmed the diagnosis of *E. vermicularis* worms [31].

## Molecular analysis

**Nested PCR results.** DNA was extracted from FFPE sections; however, due to the old storage of blocks, possible DNA fragmentation, and a little remnant of the parasite in the tissue, the results were negative in the outer PCR with CO1 primers, so the inner primer EVF/R used for successful amplification.

Of the 17 positive FFPE samples, twelve (70.6%) cases produced a band of 378 bp after nested PCR, whereas five (29.4%) samples did not have the visible product after reamplification (Fig 2). The oldest cases with positive nested PCR were from 1998.

A total of twelve amplicons of nested PCR related to FFPE tissue were subjected to sequencing. Also, the adult worm of *E. vermicularis* preserved in alcohol 70% successfully was

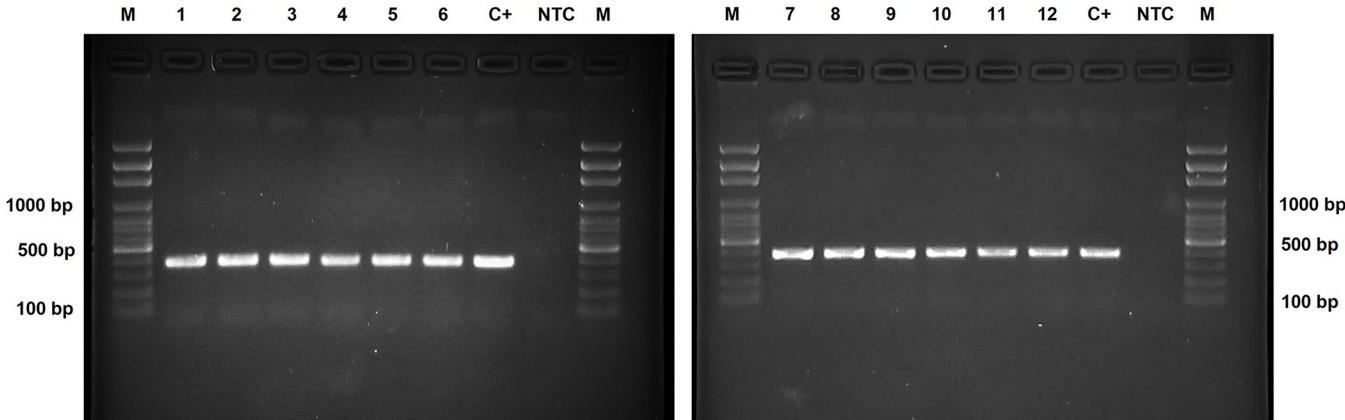

**Fig 1.** Microscopic images of the cross-section of *E. vermicularis* in the appendix lumen in different magnifications (A, B, and C). Arrows illustrate the cephalic alae located bilateral and symmetrically in the sections. Images were drawn with camera lucida, *Enterobius vermicularis* cross-section in tissue sections emphasizing cephalic alae (D, E, and F). C1: Flattened and less than five somatic muscles in each quadrant in cross-section indicated platymyarian and meromyarian types of the somatic muscle.

**Fig 2. Electrophoresis of *cox*1 gene (378 bp) amplified fragments using nested PCR in paraffin tissue samples of patients with appendicitis due to *E. vermicularis* in Tehran and Hamedan provinces.** M:100 bp DNA Marker (Gene Ruler™, Fermentas); lane numbers 1–12: amplicons related to FFPE tissues; C +: positive control (the adult worm); NTC: no template control negative control.

amplified in the first run of PCR with outer primers (CO1), and the amplicon size of 439 bp was sequenced.

## Genetic variation in the *cox*1 gene of *Enterobius vermicularis* from the appendectomy

Twelve sequences were deposited in GenBank with accession nos. MZ361991-9, MZ360956-8 (Table 1, S1 Table). In addition, the sequence of an adult worm, *E. vermicularis*, isolated from a patient in Mazandaran Province, Iran, was also deposited in GenBank with accession no. MZ362434. The fragment of 378 bp was considered for determining genetic variation between the isolates and sequences in GenBank. It should be mentioned that the length of one sequence related to the Hamedan block after several attempts was concise (277 bp), which was not considered in the genetic variation and phylogenetic analysis.

Twelve sequences (eleven from blocks and one from the adult worm), each with 378 bp in the *cox*1 gene, were analyzed for genetic variation (S1 Fig). All the sequence has the highest homology with *Enterobius vermicularis*. Median-Joining haplotype networks analysis demonstrated 12 haplotype sites (3.2%), leading to the characterization of 6 haplotypes among the sequences. Seven sequences, including five from Hamedan, and two from Tehran, represent a single haplotype. Whereas four sequences of MZ361999 (Hamedan), MZ361996 (Hamedan), MZ361995 (Hamedan), MZ361958 (Tehran), and MZ362434 (Mazandaran) each represent a single haplotype (Fig 3). Using DNAsp, when two provinces were compared, 2 and 4 haplotypes were recognized among three sequences from Tehran and eight sequences from

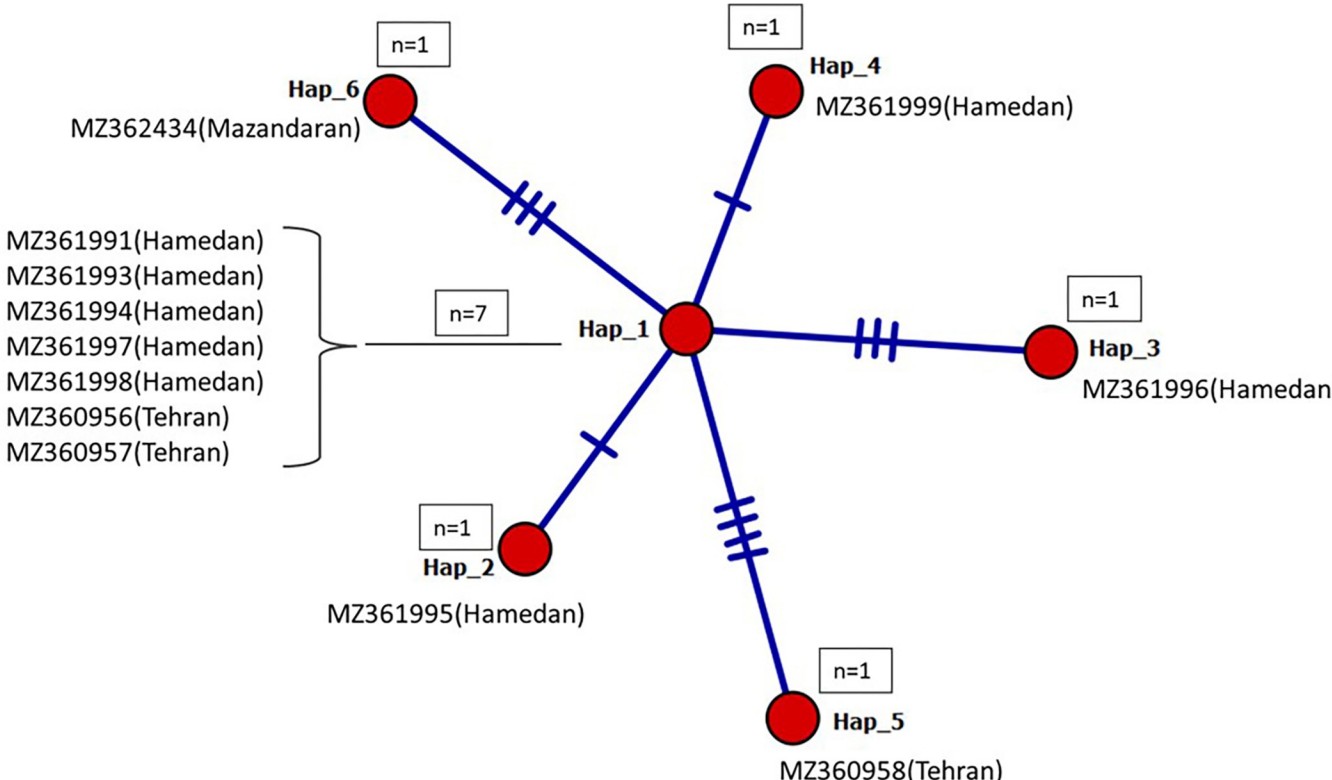

**Fig 3. Median-Joining haplotype networks analysis of *cox*1 gene (378 bp) based on 12 isolates of *E. vermicularis* constructed using PopArt 1.7.** Six haplotypes were identified within the isolates in the current study. The details of the sequence(s) belonging to each haplotype are noted. Each circle represents a unique haplotype, n represents the number of sequences in each haplotype, and the hatch marks along the network branches indicate the number of mutations.

**Table 2. Details of statistics genetic diversity for 12 sequences of *E. vermicularis* in Iran.**

| Provinces | Length (bp) | Sample size (n) | Polymorphic sites | Number of Haplotypes | Haplotype diversity (HD ± SD) | Nucleotide diversity (π ± SD) | Theta-W (Θw ±SD) | Nucleotide composition | |
|---|---|---|---|---|---|---|---|---|---|
| Hamedan | 377 | 8 | 5 | 4 | 0.643±0.184 | 0.00332±0.00144 | 0.00512 ±0.00301 | %T | 45.2 |
| | | | | | | | | %C | 9.3 |
| | | | | | | | | %A | 19.9 |
| | | | | | | | | %G | 25.6 |
| Tehran | 377 | 3 | 4 | 2 | 0.667±0.314 | 0.00707±0.00333 | 0.00707 ±0.00509 | %T | 45.2 |
| | | | | | | | | %C | 9.3 |
| | | | | | | | | %A | 19.9 |
| | | | | | | | | %G | 25.6 |
| [a]Mazandaran | 377 | 1 | 3 | 1 | - | - | - | %T | 45.1 |
| | | | | | | | | %C | 9.3 |
| | | | | | | | | %A | 20.2 |
| | | | | | | | | %G | 25.5 |
| Total | 377 | 12 | 12 | [b]6 | 0.682±0.148 | 0.00531±0.00183 | 0.01054 ±0.00491 | - | - |

[a]Due to the small number of sequences, it was impossible to estimate haplotype and nucleotide diversity and Theta-W indices for the Mazandaran case (access no. MZ362434), so it did not consider for the estimation individually.

[b]One haplotype is common in Hamadan and Tehran provinces.

Hamedan, respectively, in which one haplotype is common (Table 2). The haplotype diversity of each geographical region ranged from HD:0.643 to HD: 0.667. In contrast, nucleotide diversity was relatively low for each province, ranging between π:0.00332 for Hamedan patients and π:0.00707 for Tehran patients. Additionally, the mutation rate at the polymorphic sites was higher in Tehran cases (Θw: 00707) compared to Hamedan (0.00512) (Table 2). Noteworthy, the diversity indices in Tehran province were higher than in Hamadan, despite the lower number of cases.

## Investigating population structure and genetic diversity among isolates of *E. vermicularis*

The genetic distance among the studied samples ranged from zero to 0.019 (0–1.9%). The results indicated that the maximum genetic distance (1.9%) was between the sequences from Tehran (MZ360958) and Hamadan (MZ361996) and also between sequences from Tehran (MZ360958) and Mazandaran (MZ362434) (Table 3). The average evolutionary divergence over sequence was estimated according to the Kimura 2-parameter model. Tehran (0.0) and Hamedan (0.01) samples showed high and low evolutionary divergence within the isolates. Likewise, the evolutionary divergence among sequences of Hamedan, Tehran, and Mazandaran provinces was assessed based on the Kimura 2-parameter model. The highest divergence distance (0.01161) was observed between the Mazandaran and Tehran samples, and the lowest (0.00525) between the Tehran and Hamedan samples.

## Variation in amino acid sequences

Variants were found in 5 positions (4%) among the amino acid sequences. As indicated in Fig 4, this type of substitution changed the amino acid sequences, particularly in the Hamadan isolate with accession nos. MZ361996 and MZ361995 were novels. All the substitutions were transitions. There were five variable sites in 126 amino acid sequences, and *E. vermicularis*

**Table 3. Pairwise genetic distance matrix of *cox*1 sequences between 12 isolated *Enterobius vermicularis*.** Evolutionary analysis was conducted in MEGAX by using the Kimura 2-parameter model. The number of base substitutions per site from between sequences is shown.

| | | 1 | 2 | 3 | 4 | 5 | 6 | 7 | 8 | 9 | 10 | 11 | 12 |
|---|---|---|---|---|---|---|---|---|---|---|---|---|---|
| 1 | MZ361991 | | | | | | | | | | | | |
| 2 | MZ361993 | 0.000 | | | | | | | | | | | |
| 3 | MZ361994 | 0.000 | 0.000 | | | | | | | | | | |
| 4 | MZ361995 | 0.003 | 0.003 | 0.003 | | | | | | | | | |
| 5 | MZ361996 | 0.008 | 0.008 | 0.008 | 0.011 | | | | | | | | |
| 6 | MZ361997 | 0.000 | 0.000 | 0.000 | 0.003 | 0.008 | | | | | | | |
| 7 | MZ361998 | 0.000 | 0.000 | 0.000 | 0.003 | 0.008 | 0.000 | | | | | | |
| 8 | MZ361999 | 0.003 | 0.003 | 0.003 | 0.005 | 0.011 | 0.003 | 0.003 | | | | | |
| 9 | MZ360956 | 0.000 | 0.000 | 0.000 | 0.003 | 0.008 | 0.000 | 0.000 | 0.003 | | | | |
| 10 | MZ360957 | 0.000 | 0.000 | 0.000 | 0.003 | 0.008 | 0.000 | 0.000 | 0.003 | 0.000 | | | |
| 11 | MZ360958 | 0.011 | 0.011 | 0.011 | 0.013 | 0.019 | 0.011 | 0.011 | 0.013 | 0.011 | 0.011 | | |
| 12 | MZ362434 | 0.008 | 0.008 | 0.008 | 0.011 | 0.016 | 0.008 | 0.008 | 0.011 | 0.008 | 0.008 | 0.019 | |

isolate from Hamedan with accession number MZ361996 with three variable sites is the most variable isolate investigated (Fig 4). In the sequence with accession number 96, two amino acids in positions 39 and 41 are new in type B. In this sequence, the amino acids alanine (A) and isoleucine (I) have been converted to threonine (T), and in both cases, the hydrophobic amino acid has been converted to neutral polar. Amino acid position 38 in the sequence with accession number 95 is also new in type B. In the sequence with accession number 95, the aromatic amino acid tyrosine (Y) has been converted to cysteine (C), which is again neutral polar. These three mutations are new in *E. vermicularis* type B, but the other changes were previously detected in the Genbank sequences. All new substitutions in the sequence of amino acids were non-synonymous substitutions and transitions in the second position of the codon. The highest haplotype diversity in Iranian isolates was mainly due to synonymous substitutions. Amino acid sequences are not different except in two samples where non-synonymous substitutions occurred.

## Phylogenetic tree

Sequences were aligned and trimmed to 333 bp according to previously published *cox*1 sequences [15, 25]. Therefore, the shorter sequences of 333 bp were not considered in molecular studies and phylogenetic trees. Maximum likelihood, neighbor-joining, and UPGMA analysis of mt-DNA *cox*1 data generated the trees with similar topology and equal levels of node support with slightly lower bootstrap values. The phylogenetic trees showed that the genus *Enterobius* at first diverged from the common ancestor of *T. microon* and *T. atelis*, then the ancestors of *E. anthropopitheci* and *E. vermicularis* were separated (Fig 5). Phylogenetic

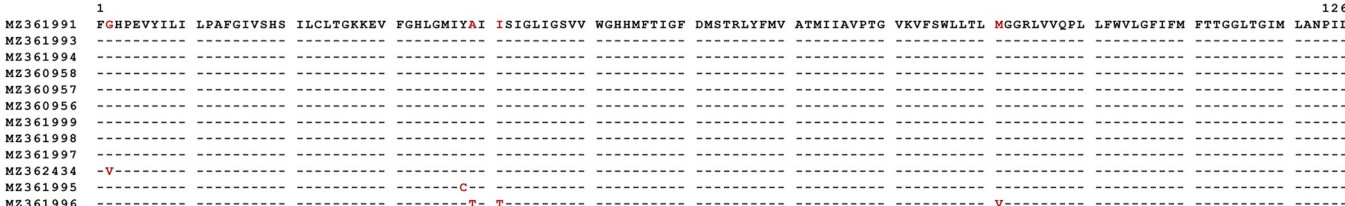

**Fig 4. The amino acid alignment of mitochondrial cytochrome c oxidase 1 (*cox*1) gene of Iranian *E. vermicularis* haplotypes with marked changes in the amino acids sequence.** Dots denote identical amino acids to the top sequence.

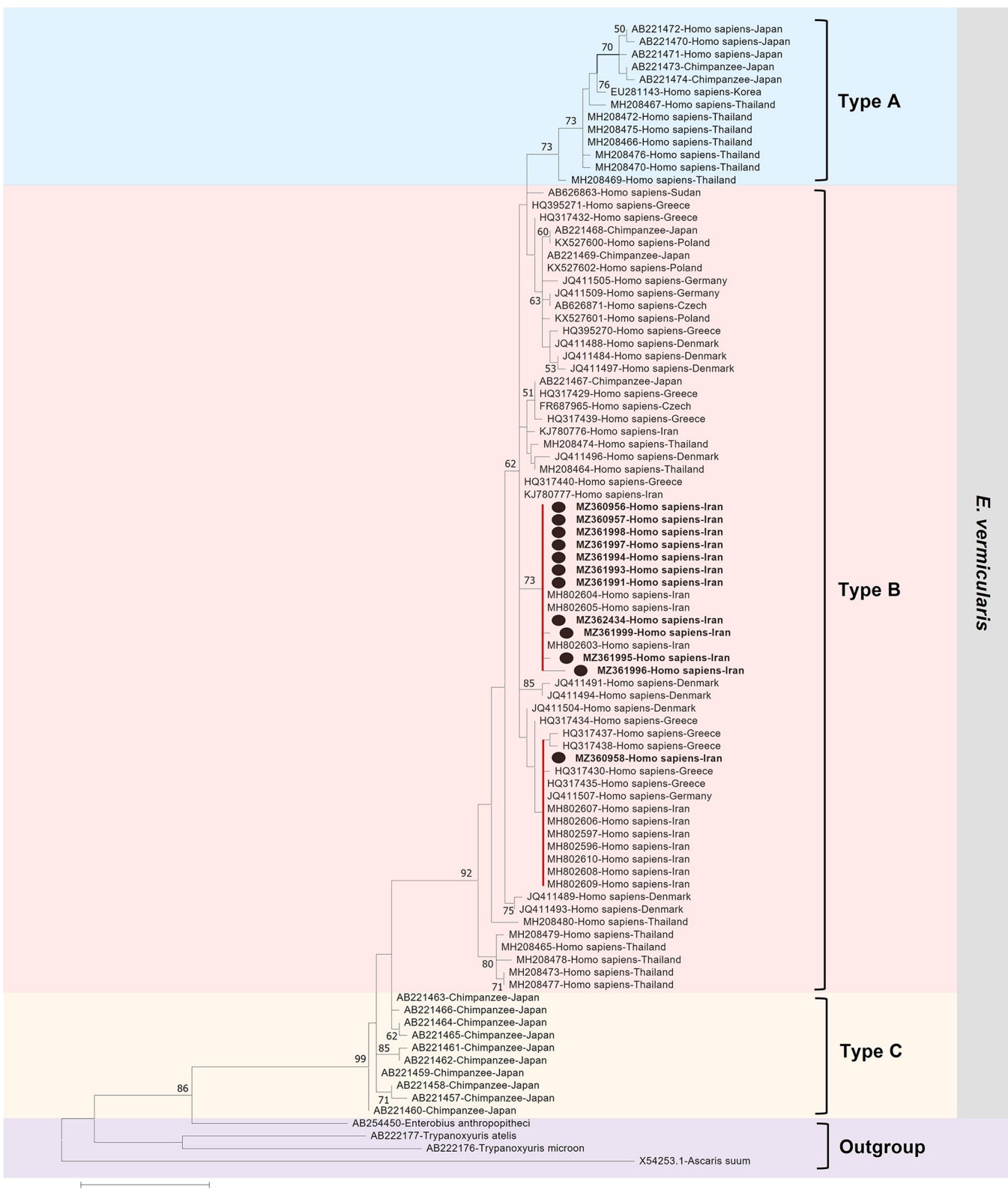

**Fig 5. Phylogenetic relationships were inferred from 333 bp of *cox1* gene sequences of *Enterobius vermicularis* isolates in this study (black circles) and others retrieved from GenBank.** The tree was constructed based on the maximum likelihood method in MEGAX with *Ascaris suum* as the outgroup.

Bootstrap values (>50%) are shown at the nodes based on 1000 replicates. The scale bar indicates genetic distance. All the reference sequences of *Enterobius vermicularis* from GenBank are displayed with their accession number, the host origins, and the original countries. Types of *E. vermicularis* and the outgroup are illustrated in different colors. The studied sequences are bold, with black circles, and placed in two clades, colored red.

analysis based on 74 *cox*1 sequences (twelve from this study and others among the available sequences of GenBank) demonstrated that the *Enterobius vermicularis* divided into three groups like former studies, with predominancy of type B (S2 Table) [15]. As shown in Fig 5, all sequences from this study are clustered within type B. All the isolates from appendectomy blocks and the adult worm were placed in the same clade with reference sequences from Iran except for MZ360958, which was placed in a separate clade with reference sequences from Greece, Germany, and Iran.

Based on the host, Type B in Iran, Greece, Germany, Thailand, Denmark, Sudan, Czech Republic, and Poland was isolated from humans, while in Japan, it was isolated just from chimpanzees. On the other hand, type A in Japan was isolated from humans and chimpanzees, while in South Korea and Thailand, human cases were just reported, and type C was just reported from chimpanzees in Japan (Fig 5, S2 Table).

### Haplotype network analysis

In agreement with the phylogenetic tree, all 12 sequences fell into type B and represented five haplotypes (Hap32, Hap37, Hap57, Hap58, Hap59). Three haplotypes, Hap57, Hap58, and Hap59, are novel, and each represented a single sequence; these haplotypes belonged to the Hamden province whose accession nos. are MZ361995, MZ361996, and MZ361999, respectively. On the other hand, Hap32 (MZ360958) and Hap37 (8 remaining sequences) were identical to a previously isolated *E. vermicularis* from Iran, Greece, and Germany (Fig 6).

No sequences from Iran fell into types A and C. As shown in Fig 6, 59 haplotypes from different countries are displayed separately as colored bullets. Peripheral haplotypes mainly had single-nucleotide variation (SNV) from central haplotypes. The amount of SNV is plotted as a line bar in Fig 6. Iranian haplotypes with red color were placed next to other haplotypes related to Type B and were separated from haplotypes of Type B and C.

The average evolutionary divergence within and between *E. vermicularis* haplotypes related to the current study and GenBank was estimated according to the Kimura 2-parameter model. It was demonstrated that the evolutionary divergence within the haplotypes of type B is higher than that of types A and C.

Also, the highest divergence distance was observed between Type C and Type A (0.0635), and the lowest was between Type B and Type A (0.0346) haplotypes.

### Discussion

Acute appendicitis is the most common severe abdominal pain requiring emergency surgery [32]. *E. vermicularis* represents one of the controversial etiologic agents for acute appendicitis. A recent meta-analysis reported that *E. vermicularis* had been detected in 4% of pathologic sections of appendicitis [33]. A definitive diagnosis of appendicitis is made through the presence of eggs or larvae worms in pathological examination of surgical specimens.

In the last decades, it has been suggested that paraffin-embedded tissues could be appropriate for the molecular identification of parasites. In this regard, PCR was applied to identify *Trypanosoma cruzi* in the paraffin-embedded placenta and fetal tissues of infected mice [34]. Müller et al. [35] used PCR to diagnose canine leishmaniasis in archival formalin-fixed skin biopsy specimens and suggested this method for retrospective epidemiological and taxonomic surveys. Also, it was demonstrated that PCR is sensitive enough to detect a single ascarid

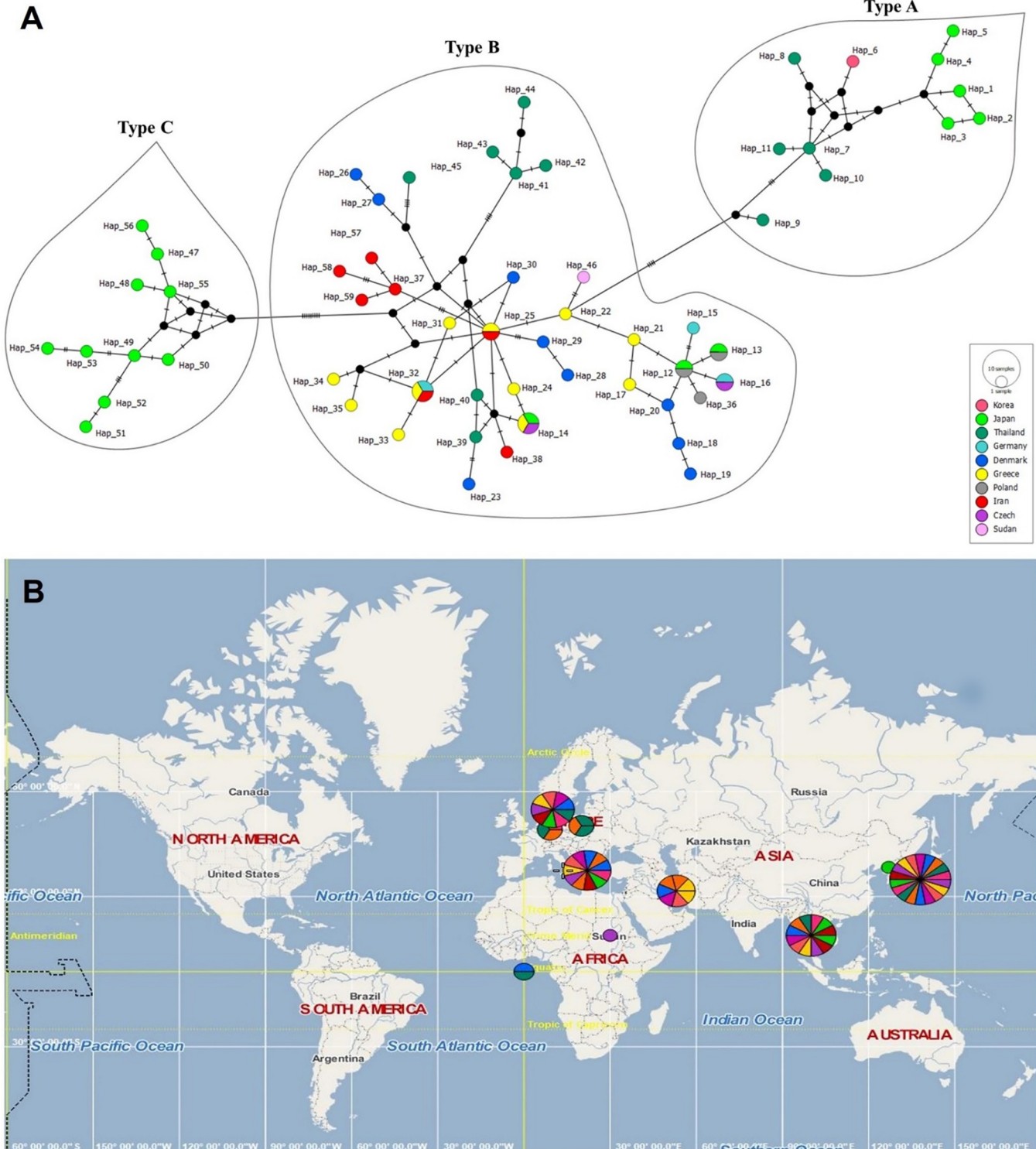

**Fig 6. Median-joining network analysis of haplotype inferred from *cox*1 sequences (333 bp) of *E. vermicularis* using PopArt 1.7 software (http://popart. otago.ac.nz).** Eighty-seven sequences were distributed among 59 haplotypes (12 sequences from this study and the rest from other studies in the NCBI database). All the sequences from this study belonged to type B. Each circle represents a haplotype, and its colors represent a country. The size of the circle indicates the relative frequency of sequences belonging to a particular haplotype. Hatch marks along the network branches indicate the number of mutations. B. Distribution and diversity of different haplotypes of *E. vermicularis* based on locations on the world map. Each color represents a different geographical location. PopArt software was used to place sample locations for sequences on the map panel (http://popart.otago.ac.nz/index.shtml) [28]. The map interface was implemented using the Marble library (http://marble.kde.org), which is distributed according to the GNU Lesser General Public License (LGPL), Version 2

nematode larvae in paraffin-embedded tissues [36]. Likewise, Rodriguez et al. evaluated PCR to identify *Angiostrongylus costaricensis* in FFPE human tissues and revealed that the molecular method has acceptable sensitivity in diagnosing abdominal angiostrongyliasis [37].

Here, for the first time, we used old-archived FFPE tissue of appendectomies for molecular characterization of *E. vermicularis* from two different geographical areas in Iran. Moreover, pathological tissue changes, morphological characteristics, and genetic variation of *E. vermicularis* were precisely studied. In the present study, eleven cases (64.71%) of appendicitis were observed in the age group below 20 years and six cases (35.29%) in the group above 20 years (Table 1). The highest number of appendicitis samples was related to the age group under 20 years old, which agrees with the prevalence of this disease at younger ages. Also, with ten cases (58.8%) out of seventeen, female was the predominant gender of appendicitis due to *E. vermicularis*. According to several studies, the prevalence rate of *E. vermicularis* in females was higher than in males, which may reflect differences in gender-related contact behavior. Indeed, female homemakers have close contact with raw vegetables, making them more susceptible to being infected with parasite eggs [10, 33].

In the current study, each FFPE sample was examined to determine histopathological changes, in which tissue changes were detected in thirteen cases (76.4%), while inflammation was observed only in four cases (23.5%). It is worth mentioning that in four specimens, despite having been diagnosed as appendicitis due to the presence of *E. vermicularis*, no inflammation and tissue changes were detected. This notion emphasizes the possible role of *E. vermicularis* in non-inflammatory appendicitis. Our pathological study revealed lymph follicle hyperplasia in the submucosal layer with no sign of classical acute appendicitis, which could be debated pathologically. In appendicitis caused by *E. vermicularis*, various inflammatory patterns such as lymphoid hyperplasia and eosinophilic or neutrophilic infiltrate can be detected [38, 39]. However, in line with our results, reactive follicular hyperplasia without inflammation was the most prevalent pathological finding in the most reported cases [5, 33, 40]. Therefore, it seems that *E. vermicularis* is not directly involved in inflammatory appendicitis due to the lack of mucosal invasion. According to several studies, clinical symptoms of appendicitis appear to be caused by obstructing the appendix lumen by the accidental entrance of the adult worm rather than actual inflammation. However, the controversial role of this parasite in acute appendicitis should not be neglected [33, 40].

In the present study, *Enterobius vermicularis* was morphologically confirmed in the cross-sections mainly based on their bilateral cephalic alae and the platymyarian and meromyarian types of somatic muscles [39]. There were no morphological differences between the cross-section of remnant *Enterobius vermicularis* in different appendectomy sections.

Molecular techniques have recently emerged as reliable identification tools to confirm microscopical parasitology. Therefore, nested PCR was administrated to amplify *E. vermicularis* cox1-gene from the old-archived appendix FFPE tissue in the current study. Archival FFPE tissues are the primary source of retrospective molecular studies; however, DNA extraction from them can be challenging because of the small size of the tissue and the insufficient quantities of DNA. Also, the cross-linking reaction of formaldehyde with DNA and protein could induce chemical modifications, DNA degradation, and fragmentation.

Furthermore, formalin can inhibit the proteinase K in the DNA extraction procedure and can interfere with PCR reaction by inhibiting DNA polymerase [41, 42]. Considering the old storage of blocks and the limitation of FFEP tissues, we used nested PCR with higher

sensitivity. Of the seventeenth FFEP samples, twelve (70.6%) cases were amplified successfully by nested PCR (Fig 2); the earliest positive cases dated back to 1998. In five blocks, nested PCR was negative, possibly due to DNA fragmentation, extended storage, or the small amount of tissue remaining in the blocks. Owing to the high DNA fragmentation, successful amplification of DNA needs shorter amplification products, 250 to 300 bp [42]; however, a higher amplicon length was used to do appropriate phylogenetic analysis. In a survey by Rodriguez et al. [37], FFPE was used for molecular diagnosis of abdominal Angiostrongyliasis, in which 50% of confirmed pathological cases were positive by PCR. The promising results of nested PCR for detecting *E. vermicularis* in appendectomy tissues can be suggested this method as a complement to microscopy. Besides FFPE tissues, we used adult worms of *E. vermicularis* from 20 years ago, over-fixed in formalin; nonetheless, due to DNA fragmentation was failed to amplify.

Genetic diversity within the *cox*1 sequence (378 bp) of twelve isolates revealed six haplotypes. Seven sequences (five from Hamedan and two from Tehran) represent a single haplotype, proposing a common source of infection. In contrast, the other five sequences each represent a single haplotype (Table 2, Fig 3). Although fewer cases belonged to Tehran, the nucleotide diversity within the sequences was higher than that of Hamadan. This finding could result from the higher diversity of patients from different geographical parts of Iran referred to the medical center in Tehran, Iran's capital city. On the other hand, the lower variation was seen within the isolates from Hamadan, where the patients are more indigenous, and less population mobility was seen.

The genetic distance among the samples varies from zero to 0.019. In this line, the intraspecies variation among the Iranian population of *E. vermicularis* eggs was 0–1.2% in the Shiraz and Khorram Abad provinces of Iran [18]. In another study, the genetic difference among *E. vermicularis* isolates in European countries was reported as 1.2% to 2.1% [16]. Also, the highest divergence distance was observed between Mazandaran and Tehran (0.01161) samples and the lowest between Tehran and Hamedan (0.00525) samples, which could be related to the different geographical distances of the origin of isolates. This association between genetic diversity and geographical distance was also reported in other *E. vermicularis* population studies in Denmark, Greece, and Thailand [16, 19, 25]. In the *Enterobius* population from Denmark, the percentage of sequence difference (p-distance) increased gradually with the geographical distance by comparing Danish isolates with the German, Greek and Japanese isolates, 1.2%, 1.5%, and 4.2%, respectively. Thus, since *E. vermicularis* is mainly specific to humans, the genetic diversity between isolates of the same region or different countries varies depending on the geographical distance and population movement [16]. Also, in the present study, the adult worm sample from Mazandaran exhibited the highest genetic distance (1.9%) compared to Tehran isolates; however, more samples should be studied to clarify the potential differences between the *E. vermicularis* isolates induced appendicitis and the other isolates inhabited in the lumen of the colon.

In the phylogenetic tree and haplotype analysis of 87 cox1 sequences (333 bp), *E. vermicularis* was classified into three groups, A, B, and C, similar to the former studies [15, 43]. Based on the differences in genetic diversity, the sequences of *E. vermicularis* were separated grossly according to the types defined by Nakano et al. [15]. Type B is the most prevalent in the human population and is predominant in Europe and Asia. It has been reported in humans in Iran, Greece, Germany, Thailand, Denmark, Sudan, the Czech Republic, and Poland, whereas it was isolated from chimpanzees in Japan [15–17, 25]. It was hypothesized that type B is probably specific to humans and possibly transmitted to captive chimpanzees from humans [15, 43]. On the other hand, type A is less prevalent and was isolated from humans and

chimpanzees in Japan and humans in Korea and Thailand [15, 19]; and Type C was just found in captive chimpanzees in Japan [15] (Figs 5, 6, S2 Table).

Our results demonstrated that the average evolutionary divergence within the haplotypes of type B is higher than that of types A and C. Also, the highest divergence distance was observed between Types C and A (%6.35), and the lowest was between Types B and A (% 3.46) haplotypes. Likewise, in a study by Nakano et al. a considerable distance of 1%, 4.1%, and 5.4% were detected for *E. vermicularis* types B, A, and C, respectively [15]. Furthermore, it was reported that the p-distances between type A and C were in the range of 3.4%-5.8%, close to 6%, which is the maximum range observed within the helminth species [13]. Since type C has just been isolated from captive chimpanzees in Japan and has the highest diversity compared to other genotypes, analysis of different target genes is needed to clarify the evolutionary status of this cryptic genotype [15]. Also, given that *cox*1 sequences of *E. vermicularis* have been published from only a limited number of countries, more studies from different geographical regions from human and captive chimpanzees are needed to explore haplotypes circulating between populations and to shed light on the potential zoonotic cycle of *E. vermicularis*.

In phylogenetic analysis, all 12 sequences from this study were clustered in the type B category and were grouped in the same clade with previously reported sequences from Iran, Khorramabad province [18, 44]. This result could be suggested the common source of infection and similar evolutionary divergence of *E. vermicularis* in Iran. On the other hand, the sequence from Tehran (MZ360958) was placed in a separate clade with sequences from Greece, Germany, and Iran (Fig 5). This similarity could be related to the high movement of the infected human population between these geographic regions [25].

Haplotypes network analysis of 87 *cox*1 sequences (333 bp) from different countries revealed 59 haplotypes, which were distributed based on *E. vermicularis*, geographical region, and population motility [16, 19]. In this network, five haplotypes were identified among the isolates compared with the GenBank sequences. Three haplotypes, including Hap57 (MZ361995), Hap58 (MZ361996), and Hap59 (MZ361999), are novel in type B and separate in different nodes where each was represented by a single sequence and belonged to Hamden province. While two other haplotypes, Hap32 (MZ360958) and Hap37 (8 remaining sequences), were similar to the *E. vermicularis* population from Iran, Greece, and Germany (Fig 6), suggesting the involvement of population mobility.

Although histological and morphological characters are not discriminative among *E. vermicularis* isolates recognized in FFPE appendectomies blocks, some limited nucleotide diversity was determined, and three new haplotypes were identified. According to the country's diverse geographical and climatological conditions, population movement, and tribe seasonal migrations, much more extensive geographical sampling is required to illustrate a more informative phylogenetic pattern within *E. vermicularis* in the human population.

## Conclusion

In conclusion, we demonstrated that old-archived FFPE tissue blocks are valuable material for molecular detection and phylogenetic analysis of *E. vermicularis* in appendectomy specimens despite the old storage, inadequate remnant tissue, DNA fragmentation, and various inhibitors. Indeed, FFPE tissue archives in the pathology department offer an underexploited resource for retrospective molecular identification and evolutionary divergence of various clinically important pathogens. Moreover, the successful utilization of nested PCR on FFPE samples points to the need for further studies to explore the diagnostic performance of this technique in a prospective study based on routine pathology. In addition, here we also

explored the pathological changes in archival FFPE appendectomies, indicating the controversial association of *E. vermicularis* with non-inflammatory appendicitis. All the *E. vermicularis* isolated from appendectomies were identified as Type B using *cox*1 sequencing; however, more representatives from different geographical regions are needed to explore the possible genotype variations between the isolates found in appendicitis.

## Supporting information

**S1 Table. The result of the identity and coverage of the sequences obtained from the *cox*1 gene of this study with the isolates of the World GenBank.**
(PDF)

**S2 Table. Sequences used in phylogeny tree, polymorphism, and haplotype network analysis according to haplotype cluster, host, and country.**
(PDF)

**S1 Fig. Alignment of 378 bp of the nucleotide sequence of the *cox*1 gene fragment isolated from the paraffin blocks obtained from this study.** The presence of the asterisk indicates similarity, and its absence indicates the difference in the sequences in that position.
(PDF)

**S1 Raw images.**
(PDF)

## Acknowledgments

We would like to acknowledge all colleagues of the pathology clinical Laboratory of Besat Hospital, Hamedan, Iran, and Children Medical Center Hospital, Tehran, Iran, for assistance in handling the samples. We thank Narges Anasori for helping with the histopathological processes.

## Author Contributions

**Conceptualization:** Elham Kazemirad, Bahram Nikmanesh, Gholamreza Mowlavi.

**Data curation:** Maryam Haghshenas, Alireza Latifi, Arash Dehghan.

**Formal analysis:** Mona Koosha.

**Funding acquisition:** Gholamreza Mowlavi.

**Investigation:** Elham Kazemirad, Gholamreza Mowlavi.

**Methodology:** Maryam Haghshenas, Mona Koosha.

**Supervision:** Elham Kazemirad, Gholamreza Mowlavi.

**Writing – original draft:** Mona Koosha, Alireza Latifi.

**Writing – review & editing:** Elham Kazemirad, Gholamreza Mowlavi.

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
