## [Decision Letter · Decision Letter 0]

23 Nov 2022

PONE-D-22-25921Detection of Enterobius vermicularis in archived formalin-fixed Paraffin-Embedded (FFPE) appendectomy blocks: It's potential to compare genetic variations based on mitochondrial DNA (cox1) genePLOS ONE

Dear Dr.  Kazemirad,

Thank you for submitting your manuscript to PLOS ONE. After careful consideration, we feel that it has merit but does not fully meet PLOS ONE’s publication criteria as it currently stands. Therefore, we invite you to submit a revised version of the manuscript that addresses the points raised during the review process.

We look forward to receiving your revised manuscript.

Kind regards,

Ebrahim Shokoohi

Academic Editor

PLOS ONE

Journal Requirements:

“This research has been supported by Tehran University of Medical Sciences & health Services grant no. 1400-1-211-52257.”

4. Please upload a new copy of Figure 4 as the detail is not clear. Please follow the link for more information: https://blogs.plos.org/plos/2019/06/looking-good-tips-for-creating-your-plos-figures-graphics/" https://blogs.plos.org/plos/2019/06/looking-good-tips-for-creating-your-plos-figures-graphics/

5. We note that Figure 6 in your submission contain [map/satellite] images which may be copyrighted. All PLOS content is published under the Creative Commons Attribution License (CC BY 4.0), which means that the manuscript, images, and Supporting Information files will be freely available online, and any third party is permitted to access, download, copy, distribute, and use these materials in any way, even commercially, with proper attribution. For these reasons, we cannot publish previously copyrighted maps or satellite images created using proprietary data, such as Google software (Google Maps, Street View, and Earth). For more information, see our copyright guidelines: http://journals.plos.org/plosone/s/licenses-and-copyright.

    a. You may seek permission from the original copyright holder of Figure 6 to publish the content specifically under the CC BY 4.0 license. 

Additional Editor Comments :

Dear Authors

We have received the paper entitled " Detection of Enterobius vermicularis in archived formalin-fixed Paraffin-Embedded (FFPE) appendectomy blocks: It's potential to compare genetic variations based on mitochondrial DNA (cox1) gene". Based on the evaluation, the submitted paper has such exciting data, especially on FFPE specimens which needs to be more studied nowadays. However, the major points raised by the Referee and the AE, are given below. Please address all raised issues point by point in the revised submission.

1-Why have you selected only coxi? Why not other genes included? Have you ever tested to be amplified by the PCR?

2-Figure 1, the scale should be in the same size and font.

2-Figure 2, is of low quality, and the authors mentioned 12 positive bands by PCR; why are only a few of them included?

3-Figure 4 is in low quality

4-In figure 5, the authors need to delete all ".1" from the accession numbers. Also, the worked sequences with accession numbers of the Iranian specimens should be given by the species name and in bold. You should also indicate the clade and discuss the clade with which your sequences are grouped with.

5- In discussion, you need to discuss your sequences based on the histology, morphology, and similarity or dissimilarity with the relevant sequences.

6-The references need to be adjusted in font and style with PLoS One.

7- the rest of the comments are included in the attached file.

Reviewers' comments:

Reviewer's Responses to Questions

**Comments to the Author**

1. Is the manuscript technically sound, and do the data support the conclusions?

Reviewer #1: Yes

2. Has the statistical analysis been performed appropriately and rigorously? 

Reviewer #1: Yes

3. Have the authors made all data underlying the findings in their manuscript fully available?

Reviewer #1: Yes

4. Is the manuscript presented in an intelligible fashion and written in standard English?

Reviewer #1: No

5. Review Comments to the Author

Reviewer #1: The manuscript including data of mtDNA of preserved specimens from Iran. The manuscript includes such interesting data, but it needs to be improved.

1-why authors selected coxi? Why other genes not been selected?

2-detail method of extraction from formalin fixed specimens not explained well.

3-in line 304, the author mentioned data not shown. Why? it should be presented or provided as supplementary file.

4-line 271 to 272 need to be revised, it is unclear.

5-English style needs to be improved.

6-The figure are not in high quality and should be improved.

6. PLOS authors have the option to publish the peer review history of their article (what does this mean?). If published, this will include your full peer review and any attached files.

Reviewer #1: No

---

## [Author Response · Author response to Decision Letter 0]

12 Jan 2023

Dear Prof. Ebrahim Shokoohi

Editor of PLOS ONE 6- January -2023 

Referring to your email concerning our manuscript entitled: " Detection of Enterobius vermicularis in archived formalin-fixed paraffin-embedded (FFPE) appendectomy blocks: It's potential to compare genetic variations based on mitochondrial DNA (cox1) gene," I would like to thank from your informative email and also from reviewers for their valuable and critical comments. Kindly inform you that all the following corrections and changes were done within the manuscript based on reviewers' suggestions and PLOS ONE's style. New and increased subjects have been highlighted in the manuscript. We hope the changes listed below meet your requirements. 

Also, regarding the funder's role, we state, "The funders had no role in study design, data collection, and analysis, decision to publish, or preparation of the manuscript."

Overall, I deeply appreciate your consideration regarding this case and look forward to hearing from you.

Sincerely Yours,

Elham Kazemirad,

Corresponding Author, Assistant Professor of Medical Parasitology 

Department of parasitology & Mycology, Tehran University of Medical Sciences,

Tel/Fax: +9821-88951392 

ekazemirad@yahoo.com , kazemirad@tuma.ac.ir

Editor comments:

Action/Comment: We edited the manuscript based on PLOS ONE's style.

2. We note that you have included the phrase "data not shown" in your manuscript. Unfortunately, this does not meet our data sharing requirements. PLOS does not permit references to inaccessible data. We require that authors provide all relevant data within the paper, Supporting Information files, or in an acceptable, public repository. Please add a citation to support this phrase or upload the data that corresponds with these findings to a stable repository (such as Figshare or Dryad) and provide and URLs, DOIs, or accession numbers that may be used to access these data. Or, if the data are not a core part of the research being presented in your study, we ask that you remove the phrase that refers to these data.

Action/Comment: We omitted the phrase "data not shown." We wrote the results in the text and mentioned the divergence distance between the geographical regions in the text. Also, in phylogenetic analysis using different methods, including Maximum likelihood, neighbor-joining, and UPGMA analysis, a similar topology was generated among cox1 sequences. All the sentences were highlighted in gray color.

3. Thank you for stating the following financial disclosure, "This research has been supported by Tehran University of Medical Sciences & health Services grant no. 1400-1-211-52257.”Please state what role the funders took in the study. If the funders had no role, please state: "The funders had no role in study design, data collection and analysis, decision to publish, or preparation of the manuscript."

Action/Comment: We state that "The funders had no role in study design, data collection, and analysis, decision to publish, or preparation of the manuscript."

4. Please upload a new copy of Figure 4 as the detail is not clear. 

Action/Comment: We upload the new high-qualified version of Figure 4.

5. We note that Figure 6 in your submission contain [map/satellite] images which may be copyrighted. All PLOS content is published under the Creative Commons Attribution License (CC BY 4.0), which means that the manuscript, images, and Supporting Information files will be freely available online, and any third party is permitted to access, download, copy, distribute, and use these materials in any way, even commercially, with proper attribution. For these reasons, we cannot publish previously copyrighted maps or satellite images created using proprietary data, such as Google software (Google Maps, Street View, and Earth). For more information, see our copyright guidelines: http://journals.plos.org/plosone/s/licenses-and-copyright.

Action/Comment: 

The map was displayed using the MARBLE package, and data comes from the OpenStreetMap database. 

We used PopArt software to place sample locations for sequences on the map panel (http://popart.otago.ac.nz/index.shtml) (1). The map interface was implemented using the Marble library (http://marble.kde.org), which is distributed according to the GNU Lesser General Public License (LGPL), Version 2 (http://www.gnu.org/licenses/lgpl.html). Map data is copyright OpenStreetMap, and is used under the Open Database License (www.opendatacommons.org/licenses/odbl ).

The fig caption was edited and highlighted in yellow.

Additional Editor Comments:

1-Why have you selected only cox1? Why not other genes included? Have you ever tested to be amplified by the PCR?

Action/Comment: Thank you for raising this point.

Cytochrome c oxidase subunit 1 (cox1) gene is one of the most diverse mitochondrial genes used for the phylogenetic analysis of E. vermicularis. It can provide a more comprehensive representation of E. vermicularis evolution than other genes like ribosomal DNA (2, 3). In the previous studies in Japan and Thailand, partial sequences of ITS2 and Cytochrome c oxidase subunit 1 (cox1) genes were compared within Enterobius vermicularis isolates and demonstrated that ITS2 has a high similarity and low nucleotide variation to use in the phylogenetic tree. On the other hand, the mitochondrial cox1 gene revealed considerable genetic diversity in phylogenetic analysis. Consequently, the cox1 gene has been proposed as a discriminative marker for genetic diversity. Based on that, E. vermicularis is divided into three groups A, B, and C (4, 5). In most studies, the genetic diversity of E. vermicularis has been demonstrated according to the cox1 gene (6-8). In this line, we used this gene to demonstrate the potential genetic variation between the isolates extracted from archived FFPE samples and cox1 sequences deposited in the GenBank.

In this regard, we edited the introduction and emphasized more on the advantage of this gene for phylogenetic analysis. (Written in purple color)

2-Figure 1, the scale should be in the same size and font.

Action/Comment: We edited the Figure.

2-Figure 2, is of low quality, and the authors mentioned 12 positive bands by PCR; why are only a few of them included?

Action/Comment: As all PCR products were subjected to sequencing, we did PCR and gel electrophoresis again. The 1/100 dilution of DNA was used to eliminate the smears. We replaced the Figure, illustrated all the bands in two different agarose gels, and also edited the caption, written in red. Also, the not-cropped gel uploaded as S1_raw_images.

3-Figure 4 is in low quality 

Action/Comment: We edited the Figure. 

4-In figure 5, the authors need to delete all ".1" from the accession numbers. Also, the worked sequences with accession numbers of the Iranian specimens should be given by the species name and in bold. You should also indicate the clade and discuss the clade with which your sequences are grouped with.

Action/Comment: In Figure five, ".1" was deleted from the accession numbers, and the specimens used in the present study were bolded. We separated the groups in the Figure by different colors and highlighted the clades in which our isolates were placed. Also, we discussed other sequences that grouped with our samples. (Highlighted in turquoise color)

5- In discussion, you need to discuss your sequences based on the histology, morphology, and similarity or dissimilarity with the relevant sequences.

Action/Comment: Although histological and morphological characters are not discriminative among E. vermicularis isolates identified in FFPE appendectomies blocks, some limited variation was clarified within the isolates. In the present study, three new haplotypes were identified in type B, and also one sequence from Tehran (MZ360958) was placed in a separate clade with sequences from Greece, Germany, and Iran. According to the country's diverse geographical and climatological conditions, population movement, and tribe seasonal migration, much more extensive geographical sampling is required to illustrate a more phylogenetic informative pattern within E. vermicularis in the human population.

 We added this statement in the discussion section and highlighted it in green.

6-The references need to be adjusted in font and style with PLoS One.

Action/Comment: We edited the manuscript based on PLOS ONE style. Also, the whole manuscript was revised grammatically, with all the changes written in blue.

7- the rest of the comments are included in the attached file.

Action/Comment: All the remaining comments were replied to in the manuscript.

Reviewers' comments:

Reviewer's Responses to Questions

Comments to the Author

1. Is the manuscript technically sound, and do the data support the conclusions?

Reviewer #1: Yes

2. Has the statistical analysis been performed appropriately and rigorously?

Reviewer #1: Yes

3. Have the authors made all data underlying the findings in their manuscript fully available?

Reviewer #1: Yes

4. Is the manuscript presented in an intelligible fashion and written in standard English?

Reviewer #1: No

5. Review Comments to the Author

Please use the space provided to explain your answers to the questions above. You may also include additional comments for the Author, including concerns about dual publication, research ethics, or publication ethics. (Please upload your review as an attachment if it exceeds 20,000 characters)

Reviewer #1: The manuscript including data of mtDNA of preserved specimens from Iran. The manuscript includes such interesting data, but it needs to be improved.

1-why authors selected coxi? Why other genes not been selected?

Action/Comment: Thanks for the precise question.

Cytochrome c oxidase subunit 1 (cox1) gene is one of the most diverse mitochondrial genes used for the phylogenetic analysis of E. vermicularis. It can provide a more comprehensive representation of E. vermicularis evolution than other genes like ribosomal DNA (2, 3). In the previous studies in Japan and Thailand, partial sequences of ITS2 and Cytochrome c oxidase subunit 1 (cox1) genes were compared within Enterobius vermicularis isolates and demonstrated that ITS2 has a high similarity and low nucleotide variation to use in the phylogenetic tree. On the other hand, the mitochondrial cox1 gene revealed considerable genetic diversity in phylogenetic analysis. Consequently, the cox1 gene has been proposed as a discriminative marker for genetic diversity. Based on that, E. vermicularis is divided into three groups A, B, and C (4, 5). In most studies, the genetic diversity of E. vermicularis has been demonstrated according to the cox1 gene (6-8). In this line, we used this gene to demonstrate the potential genetic variation between the isolates extracted from archived FFPE samples and cox1 sequences deposited in the GenBank.

In this regard, we edited the introduction and emphasized more on the advantage of this gene for phylogenetic analysis. (Written in purple color)

2-detail method of extraction from formalin fixed specimens not explained well.

Action/Comment: We explained the detail of DNA extraction from FFPE tissue, highlighted in pink.

3-in line 304, the Author mentioned data not shown. Why? it should be presented or provided as supplementary file.

Action/Comment: We omitted the phrase "data not shown." We wrote the results in the text and mentioned the divergence distance between the geographical regions in the text. Also, in phylogenetic analysis using different methods, including Maximum likelihood, neighbor-joining, and UPGMA analysis, a similar topology was generated among cox1 sequences. All the sentences were highlighted in gray color.

4-line 271 to 272 need to be revised, it is unclear.

Action/Comment: It was revised and written in orange color.

5-English style needs to be improved.

Action/Comment: It was edited, and all the new editions and changes were written in blue.

6-The Figure are not in high quality and should be improved.

Action/Comment: All the figures were improved and replaced.

6. PLOS authors have the option to publish the peer review history of their article (what does this mean?). If published, this will include your full peer review and any attached files.

Do you want your identity to be public for this peer review? For information about this choice, including consent withdrawal, please see our Privacy Policy.

Reviewer #1: No

References

1. Leigh JW, Bryant D. POPART: full-feature software for haplotype network construction. Methods Ecol Evol. 2015; 6:1110-6.

2. Blouin MS. Molecular prospecting for cryptic species of nematodes: mitochondrial DNA versus internal transcribed spacer. Int J Parasitol. 2002; 32:527-31. https://doi.org/10.1016/s0020-7519(01)00357-5 PMID: 11943225. 

3. Kang S, Sultana T, Eom KS, Park YC, Soonthornpong N, Nadler SA, Park JK. The mitochondrial genome sequence of Enterobius vermicularis (Nematoda: Oxyurida)--an idiosyncratic gene order and phylogenetic information for chromadorean nematodes. Gene. 2009; 429:87-97. https://doi.org/10.1016/j.gene.2008.09.011 PMID: 18848867.

4. Nakano T, Okamoto M, Ikeda Y, Hasegawa H. Mitochondrial cytochrome c oxidase subunit 1 gene and nuclear rDNA regions of Enterobius vermicularis parasitic in captive chimpanzees with special reference to its relationship with pinworms in humans. Parasitol Res. 2006; 100:51-7. https://doi.org/10.1007/s00436-006-0238-4 PMID: 16788831.

5. Tomanakan K, Sanpool O, Chamavit P, Lulitanond V, Intapan PM, Maleewong W. Genetic variation of Enterobius vermicularis among schoolchildren in Thailand. J Helminthol. 2018; 94:e7. https://doi.org/10.1017/s0022149x18000962 PMID: 30369341.

6. Ferrero MR, Röser D, Nielsen HV, Olsen A, Nejsum P. Genetic variation in mitochondrial DNA among Enterobius vermicularis in Denmark. Parasitology. 2013; 140:109-14. https://doi.org/10.1017/s0031182012001308 PMID: 22906211.

7. Kubiak K, Dzika E. Enterobiasis epidemiology and molecular characterization of Enterobius vermicularis in healthy children in north-eastern Poland. Helminthologia. 2017; 54:284-91.

8. Hagh VR, Oskouei MM, Bazmani A, Miahipour A, Mirsamadi N. Genetic classification and differentiation of Enterobius vermicularis based on mitochondrial cytochrome c oxidase (cox1) in northwest of Iran. J Pure Appl Microbiol. 2014; 8:3995-9.

---

## [Decision Letter · Decision Letter 1]

30 Jan 2023

Detection of Enterobius vermicularis in archived formalin-fixed paraffin-embedded (FFPE) appendectomy blocks: It's potential to compare genetic variations based on mitochondrial DNA (cox1) gene

PONE-D-22-25921R1

Dear Dr. Kazemirad,

We’re pleased to inform you that your manuscript has been judged scientifically suitable for publication and will be formally accepted for publication once it meets all outstanding technical requirements.

Kind regards,

Ebrahim Shokoohi

Academic Editor

PLOS ONE

Additional Editor Comments (optional):

The paper has been improved. The concerns raised by the AE and Referee were addressed.

Reviewers' comments:

Reviewer's Responses to Questions

**Comments to the Author**

1. If the authors have adequately addressed your comments raised in a previous round of review and you feel that this manuscript is now acceptable for publication, you may indicate that here to bypass the “Comments to the Author” section, enter your conflict of interest statement in the “Confidential to Editor” section, and submit your "Accept" recommendation.

Reviewer #1: All comments have been addressed

2. Is the manuscript technically sound, and do the data support the conclusions?

Reviewer #1: Yes

3. Has the statistical analysis been performed appropriately and rigorously? 

Reviewer #1: Yes

4. Have the authors made all data underlying the findings in their manuscript fully available?

Reviewer #1: Yes

5. Is the manuscript presented in an intelligible fashion and written in standard English?

Reviewer #1: Yes

6. Review Comments to the Author

Reviewer #1: All comments and questions raised were addressed and the authors improved the paper. This paper has interesting information on the parasite and as the authors worked on the mtDNA as one of the important marker in this filed, the paper provide good information for the researchers and all readers interested.

7. PLOS authors have the option to publish the peer review history of their article (what does this mean?). If published, this will include your full peer review and any attached files.

Reviewer #1: No

---

## [Editor Report · Acceptance letter]

1 Feb 2023

PONE-D-22-25921R1 

Detection of *Enterobius vermicularis* in archived formalin-fixed paraffin-embedded (FFPE) appendectomy blocks: It's potential to compare genetic variations based on mitochondrial DNA (*cox*1) gene 

Dear Dr. Kazemirad:

I'm pleased to inform you that your manuscript has been deemed suitable for publication in PLOS ONE. Congratulations! Your manuscript is now with our production department. 

Kind regards, 

on behalf of

Dr. Ebrahim Shokoohi 

Academic Editor

PLOS ONE